# Process evaluation of enhancing primary health care for non-communicable disease management in Malaysia: Uncovering the fidelity & feasibility elements

**Lee Lan Low**[1]*, **Fathullah Iqbal A. B. Rahim**[1], **Nur Aqlili Riana Hamzah**[1], **Mohd Safiee Ismail**[2]

1 Institute for Health Systems Research, National Institutes of Health, Ministry of Health Malaysia, Putrajaya, Malaysia, 2 Family Health Development Division, Ministry of Health Malaysia, Putrajaya, Malaysia

* low.ll@moh.gov.my

## Abstract

### Background

In combating the increasing trend of non-communicable diseases (NCDs) over the last two decades in the country, the Ministry of Health Malaysia developed the Enhanced Primary Health Care (EnPHC) initiative to improve care management across different levels of the public service delivery network. An evaluation research component was embedded to explore the implementation issues in terms of fidelity, feasibility, adaptation and benefit of the initiative's components which were triage, care coordination, screening, risk management and referral system.

### Methods

A mixed methods study was conducted at 20 participating EnPHC clinics in Johor and Selangor, two months after the intervention was initiated. Data collected from self-reported forms and a structured observation checklist were descriptively analysed. In-depth interviews were also conducted with 20 participants across the clinics selected to clarify any information gaps observed in each clinic, and data were thematically analysed.

### Results

Evaluation showed that all components of EnPHC intervention had been successfully implemented except for the primary triage counter and visit checklist. The challenges were mainly discovered in terms of human resource and physical structure. Although human resource was a common implementation challenge across all interventions, clinic-specific issues could still be identified. Among the adaptive measures taken were task sharing among staff and workflow modification to match the clinic's capacity. Despite the challenges, early benefits of implementation were highlighted especially in terms of service outcomes.

**Data Availability Statement:** The dataset that support the findings of this article belongs to the Enhanced Primary Healthcare Evaluation study. Request for data can be obtained from Dr. Mohd

Azahadi Omar (drazahadi@moh.gov.my), the head of sector for Biostatistics & Data Repository, National Institute of Health, Ministry of Health Malaysia and with the permission from the Director General of Health, Malaysia.

**Funding:** This study was made possible with a funding from the National Institutes of Health, Ministry of Health Malaysia research grant [(21) KKM/NIHSEC/800-3/2/1 Jld.5]. The funders had no role in study design, data collection and analysis, decision to publish, or preparation of the manuscript.

**Competing interests:** The authors have declared that no competing interests exist.

**Abbreviations:** CC, Care Coordinator; EnPHC, Enhanced Primary Health Care; EnPHC-PE, Enhanced Primary Health Care-Process Evaluation; HCP, health care provider; IDI, in-depth interview; LO, liaison officer; MOIC, medical doctor in charge; NCD, non-communicable disease; PHC clinic, primary health care clinic.

## Conclusions

The evaluation study disclosed issues of human resource and physical infrastructure when a supplementary intervention is implemented. To successfully achieve a scaled-up PHC service delivery model based on comprehensive management of NCDs patient-centred care, the adaptive measures in local clinic context highlight the importance of collaboration between good organisational process and good clinical practice and process.

## Background

Non-communicable diseases (NCDs) and their multiple risk factors have been affecting the Malaysian population, across wide age range and income level [1]. NCDs have become a burden to the healthcare systems in Malaysia where approximately 70% of the patients' treatment cost is funded by the government, as most patients sought treatment in government healthcare facilities [2,3]. A recent study has found that the diabetes treatment expenditure itself is two to three times higher as compared to non-diabetes treatment [3], and the Disability-Adjusted Life Year monetary losses from diabetes totalled RM 10.21billion [2].

Factors influencing the community's healthcare utilisation include ease of access to the primary healthcare facilities such as public clinics [4], distance to clinic [5], time apportioned to travel back and forth [6] as well as the lower fee charged at public primary healthcare as compared to private health sector treatment. Since NCDs require long-term treatment, accessibility and affordability are regarded as two key factors for NCD treatment. Hence, the public primary healthcare became the main choice by the rural and urban communities.

Due to the alarming rise of NCD burden, Ministry of Health Malaysia (MOH) through the National Strategic Plan for Non-Communicable Disease (NSP-NCD) 2016–2025 has proposed a number of initiatives including "Strengthening Chronic Disease Management at Primary Care Level through the Enhanced Primary Health Care (EnPHC) Initiative" [7]. The EnPHC initiative is a complex intervention package designed to improve NCDs care management involving community, primary care, secondary care and tertiary care, by focusing on NCDs and their risk factors [8], which are diabetes mellitus, hypertension and high cholesterol.

Box 1 describes the components of the EnPHC intervention, namely deployment of primary and secondary triage counters, introduction of the care coordinator, new NCD screening strategy, enhancing NCD risk management (risk stratification and health education) as well as improvements on the referral system (referral within clinic and to hospitals) for a better NCD care management and evidence-based nationwide scalability [9].

Evaluation research on an intervention is vital to determine feasibility of implementation in the setting and to identify early problems which occur over the implementation course. While the quantitative evaluation method is able to quantify fidelity and utilisation of the components of the intervention, to gain an understanding of how the intervention components could be translated from research into practice, the qualitative approach is taken to explore the reasoning behind its feasibility, as well as to support the quantitative findings [22,23]. This was among the MOH's first few complex interventions which had a process evaluation built into its design. A process evaluation is therefore essential to assess EnPHC's complex intervention. The aim of this process evaluation, named as EnPHC-PE, was to assess the implementation of EnPHC intervention packages, particularly on determining whether EnPHC intervention components were implemented as designed in the clinics (fidelity), feasibility as well as perceived benefits from the healthcare providers (HCPs), either professionals (medical officers

---

### Box 1: The EnPHC intervention components

#### Deployment of triage counters

The triaging system is commonly associated with a hospital's Emergency Department [10,11]. EnPHC introduced this system in clinics, in the form of primary and secondary triage counters. The primary triage counter aims to effectively channel and direct patients according to case designation (life-threatening, urgent, infectious or non-infectious) to the appropriate service point in the clinic [12]. The secondary triage counter aims to screen and detect undiagnosed NCD cases, and ensure care continuity for ongoing NCD cases.

#### Introduction of the care coordinator (CC) role

Care Coordinator is a key component introduced by EnPHC to integrate, facilitate and coordinate NCD care-related activities [13,14]. A CC manages the newly introduced NCD care form and visit checklist, defaulter tracing as well as referrals. The task and responsibility to manage patient data and patient outreach were shifted from doctors in the clinic to the CCs [15]. The NCD care form is a modified concise patient medical record for NCD care management. The visit checklist is an electronic registry capturing a patient's demographic and brief clinical information during each clinic visit, as well as current and future appointment dates to facilitate the tracing of appointment and medication refill defaulters.

#### NCD screening

The preconceived notion of "a general health screening is a waste of time unless you have symptoms" was a big challenge [16,17]. EnPHC aimed to change such a notion by conducting opportunistic NCD screening activities for non-NCD clinic attendees. This enabled NCD risk factor assessment to determine the subsequent care pathway, either annual screening at the community level or further management in the clinic.

#### NCD risk management

This activity was enhanced to empower assistant medical officers and staff nurses, which was previously only performed by doctors. It consisted of cardiovascular disease risk stratification (using the Framingham risk score [18–20]) and health education for the patients.

#### Referral systems

EnPHC introduced an internal referral system within the clinic named Integrated Specialised Services (ISS) to in-house or visiting allied health professionals such as nutritionist, physiotherapist and dietitian [21]. The external referral mechanism to specialist clinics in hospitals (such as nephrology, ophthalmology and orthopaedics) was also improved under EnPHC.

---

and pharmacist) or assistants medical officers (also known as physician's assistant, who received a 3-year training course which allows them to diagnose and treat minor ailments and to conduct minor surgical procedures) and staff nurses, regarding the implementation of EnPHC.

## Methods

### Study design and sampling

The EnPHC initiative was piloted in July 2017 in 20 public primary care clinics in two states of Johor and Selangor in Malaysia. Selection of the two states was based on the balance between regional representativeness, intervention budget and implementation capacity which involves manpower and facilities [24]. For the purpose of evaluation, an addition of 20 control clinics were selected, matched according to urban-rural stratification, clinic setting and type as well as human resource availability in the clinic. Three evaluation studies on the EnPHC initiative were conducted concurrently and independently; two outcome evaluation at the community and facility level [25,26], and process evaluation (EnPHC-PE) focusing on the implementation process [9]. Since our focus was on the implementation fidelity and feasibility, the control clinics were excluded from the process evaluation.

EnPHC-PE was conducted to explore four aspects of implementation issues, namely fidelity, feasibility, adaptation and perceived benefit. The evaluation aspects are as the following:

Fidelity is defined as the degree to which an intervention was implemented as it was prescribed in the original protocol or as it was intended by the program developers [27–29]. Fidelity was assessed from two perspectives; availability of the implemented intervention and its utilisation according to guidelines and protocols.

Feasibility is defined as the extent to which a new treatment, or an intervention, can be successfully used or carried out within a given agency or setting [30].

Adaptation is described as modifying the changes which have been tested and found to be effective and suited to local setting or condition [31].

Benefit is an advantage or profit gained. The benefits of an intervention targeting on the ultimate outcome are dependent on the effectiveness of other interventions throughout the care cycle [32].

The mixed method study design using structured observation and in-depth interview (IDI) was conducted from September to November 2017, two months after EnPHC was rolled out in the 20 participating public primary health care (PHC) clinics. As the clinic setting, infrastructure and human resource varied across the clinics, the adjustments in implementation were expected to be different. Hence, two months were deemed appropriate and sufficient period for the clinics to make the necessary adjustments.

### Data collection and participant recruitment

Data was collected using three approaches via self-reported assessment form, structured observation checklist and in-depth interviews.

The self-reported assessment form—this data collection tool consisted of three main variables; participants were required to identify the intervention components which have been implemented in their clinic from the list shown in the form, to fill in the implementation date for each intervention as well as an open-ended section to state the reasons if there were any intervention that is yet to be implemented. The form was distributed to all PHC clinics, filled by the clinic's liaison officer (LO) for EnPHC that was appointed by the program owners and collected before the structured observation activity was conducted. This initial assessment was important to understand the implementation status of each clinic which later on facilitated the development of the checklist for structured observation.

The structured observation checklist—this tool was developed to facilitate researchers' observations on how each of the interventions were implemented and carried out in the clinics. The checklist was pre-tested in one of the participating PHC clinics. Information during pre-testing was captured as field notes which was used to improve the checklist, specifically on the observations for the ISS intervention and document formatting to create more space for researchers to add notes. The revised checklist was later used for the remaining 19 clinics [Refer Appendix for the structured observation checklist]. The structured observation activity was conducted in all 20 PHC clinics, to cover all interventions that were implemented as well as to assess the implementation in the different departments within the clinic. For each clinic, three to five research team members made their observations independently, taking between 45 minutes to two hours for completion. The checklist was also designed with space for the researcher to write the field notes. To follow-up with the observation, the research team members for a particular clinic will gather and discuss among themselves for 10–30 minutes on specific issues noted during their observation which needed further exploration during the subsequent IDI sessions.

The in-depth interview (IDI)–semi-structured interviews were conducted with 20 HCPs respectively from each intervention clinic. Selection of interviewees was purposive to reflect sharing of implementation issues across clinic settings by the different categories of staff holding different roles and responsibilities. IDI aimed to explore the implementation issues and to clarify information gaps from the self-assessment form and/or the observation.

Each participant was briefed on the interview process and guaranteed the confidentiality prior to the session while obtaining informed consent and permission for audio-recording during the interview. The interview was conducted by trained research team members in the participants' preferred language either English or Malay. It was conducted on the same day as the observation with each session lasting between 30 minutes to two hours.

## Data management and analysis

Fidelity information was assessed using self-reported forms followed by a structured observation checklist which was later entered and tabulated in Microsoft Excel. The data were descriptively analysed and presented as the number of availability and utilisation of each intervention component. The qualitative data from IDI were in audio recording and transcribed verbatim. Identifiers for the interviewer and participants remained anonymous and were replaced with researcher-assigned numbers for confidentiality purposes. Thematic analysis approach was used to determine themes and sub-themes from the interview transcripts based on objectives (feasibility, adaptation and perceived benefit). The results were presented in form of excerpts.

The quality control process was carried out by research team members for both quantitative and qualitative data. The data obtained from structured observation was fully cross-checked with the raw data obtained from the checklist. Prior to analysis of qualitative data, research team members who were not involved in the transcribing process cross-checked the transcript with the audio recording for accuracy. Data interpretation was verified by contacting study participants through telephone text messages to ensure the interpreted meaning was as intended. Data validation was achieved through consensus among the multidisciplinary research team members [9]. In addition, triangulation of data collection methods and data sources was performed to ensure the in-depth understanding of implementation issues for each intervention component and whether the issue was similar across clinics. The preliminary data were then presented to the program owners and members of the intervention design team for further triangulation of findings and the intended design.

## Ethical consideration

The EnPHC-PE study was registered under the National Medical Research Register, Ministry of Health Malaysia (NMRR-17-295-34771). Ethics approval was granted by the Medical Research and Ethical Committee (MREC), Ministry of Health Malaysia (ref: (5) KKM/NIH-SEC/P17-350). Confidentiality of participants involved was assured throughout the study's conduct. Additional permission from the management of the respective state and clinic was obtained. Written informed consent from participants was obtained prior to the interview and confidentiality of participants was assured throughout the study's conduct.

## Results

In total, 16 females and four males who were interviewed involved five Medical Officer in Charge (MOICs), 12 LOs and three CCs (Table 1). The participants' age ranged between 30 to 47 years old, with a mean age of 37.68 (SD 5.4). The interventions were evaluated according to the aspects of Fidelity, Feasibility, Adaptation and Perceived Benefit.

## Fidelity

Fidelity assessment was based on the availability of the intervention and its utilisation in accordance to guideline protocols. In general, all intervention components have been implemented according to the protocols in all 20 clinics, except for the following specific components:

**Triage counters (primary triage counter).**   All clinics deployed the primary triage counter except for one clinic which failed to establish the required counter due to limitation in building infrastructure. Four primary triage counters from the remaining 19 clinics were found unattended and acted as placeholders. The secondary triage counter, however, was successfully set up and utilised in all clinics.

**Care coordinator (visit checklist).**   One clinic failed to utilise the required format of the visit checklist as they deemed the existing patient appointment card sufficient to capture the required information.

**Care Coordinator (appointment defaulter tracing).**   The appointment defaulter tracing activity was successfully conducted in all clinics. However, as there was no guideline protocol

**Table 1.  Participant demographic (n = 20).**

| Participants Characteristics | Total | Percentage (%) |
|---|---|---|
| **Sex** | | |
| Male | 4 | 20 |
| Female | 16 | 80 |
| **Participants' age (in years)** | | |
| Age range | 30–47 | |
| Mean (SD) | 37.68 (5.4) | |
| **Participants' Role in the Clinic** | | |
| Medical Officer in Charge (MOIC) | 5 | 25 |
| Liaison Officer (LO) for EnPHC; | | |
| 1. Medical Doctor | 9 | 45 |
| 2. Staff Nurse | 3 | 15 |
| Care Coordinator (CC); | | |
| 1. Assistant Medical Officers | 2 | 10 |
| 2. Staff Nurse | 1 | 5 |

Note: SD = Standard Deviation.

on how it should be conducted, fidelity assessment could not be made as there was no standard to match. This issue was cited to contribute into feasibility issues during further exploration through IDI.

**Referrals (external referral).** The communication between clinics and hospitals on referral management had been established in all clinics. However, fidelity assessment was unsuccessfully conducted on the communication process as the guideline has no specific criteria to be communicated in the referral as standards to be met.

## Feasibility

**Triage counters.** Limitation in physical infrastructure was a common issue encountered by the clinics to deploy the triage counters. Lack of physical space was especially common in clinics with an old building. When these clinics placed the triage counters inside the building, long queues were formed, thereby leading to congestion. In response to this matter, physical renovations were proposed but were deemed unfeasible due to lack of funding.

Human resource was another common challenge expressed by the participants. The EnPHC standard operating protocol outlined that the triage counters should be managed by specified categories of staff namely, assistant medical officers and staff nurses. However, they were currently multi-tasking and had existing responsibilities which influenced the intended counter's function. The primary triage counters were occasionally left unattended in the event of responsibility prioritisation such as attending to emergency cases or covering for other staffs on leave.

*"Unfortunately . . .two issues there. First is space and the second is the person who has to be at the primary triage counter, which is the MA (assistant medical officer), also limited."*

[Clinic 2, Assistant Medical Officer, 37 years old].

**Care coordinator.** The assistant medical officer and staff nurse were found competent to manage the NCD cases in accordance with the standard operating procedure, the clinical protocols as well as the EnPHC guideline which was provided for each intervention clinic. An issue raised on the design of NCD care form was the insufficient space to write referral note. The visit checklist underwent several updates, re-training was not conducted for each update. It was left to the clinic staff to adapt and implement the latest version.

Patients' non-adherence to appointments and the implementation of the appointment defaulter system was a common challenge across the clinics. As the implementation guidelines on this activity was not written in detail, the clinics adopted various mechanisms. While all the clinics performed the activity primarily by phone call, the follow-up mechanism differed. If an appointment defaulter was unreachable by phone, the clinic staff would contact his/her next of kin or the patient's neighbour. Eight of the clinics even took the extra initiative of doing home visits, but it brought in logistical challenges of tracing houses when it came to incomplete addresses.

**NCD screening.** The implementation of NCD screening was perceived as challenging due to inadequate staff, as shared by one participant: *"conducting patient screening is a barrier because we don't have enough staff"* [Clinic 9, Professional, 36 years old]. However, the activity managed to be implemented in all clinics due to the staffs' dedication, commitment and teamwork in adapting to this new task which they deemed to be valuable.

**NCD risk management.** The participants perceived that the use of the mobile applications for the Framingham risk score caused patients to wait longer. As there were frequent

application malfunctions and issues in internet access, they tried to overcome the obstacle by performing manual calculations. Participants shared that patients may have a negative impression of staff using the mobile application, perceiving that they were playing with their phones while on duty. The lack of a dedicated room while delivering health education to patients at the open counters raised concerns on patient privacy. Despite the limitations, the clinic staff attempted to conduct opportunistic health education activities, as shared by a participant: *"We conducted (health education sessions) in rotations, with pharmacy, with diabetic foot care, with doctors"* [Clinic 10, Staff Nurse, 45 years old].

**Referrals.**    The space for referral feedback in the NCD care form was very limited which posed risks for incomplete information. Another issue raised by the staff was the continued use of an existing protocol which stipulated for the existing official referral form to be filled, resulting in repetitive data entry into two separate documents for the same purpose.

## Adaptation

**Triage counters.**    The physical infrastructure was a limitation, the deployment of the triage counters caused patient congestion in most of the clinics. To overcome the problem, modifications of the primary triage counter were carried out in four clinics to also function as a registration counter for new patient visits. The setup for the secondary triage counter also varied, from actual physical counters which were stated in the implementation protocol to dedicated rooms. The rooms were viewed as necessary in eleven clinics for privacy reasons as well as providing a conducive environment for staff to carry the task, as shared by a participant: *"Patient's privacy is more secured and staffs feel more comfortable (working)"* [Clinic 4, Professional, 33 years old].

Due to the human resource constraints, all clinic staff shared the task of managing the primary triage counters instead of assigning dedicated personnel. To overcome the manpower shortage issue, a participant shared: *"Not fixed (staff)...depending on the staff availability on that day"* [Clinic 2, Assistance Medical Officer, 37 years old]. For the secondary triage, one of the clinics grouped their staff into two teams by the zone catchment area of the clinic. The schedule was modified so that one team would dedicate two days per week for NCD risk management to patients in their assigned zone, while another team took care of the general outpatients, as shared by one of the participants: *"Monday and Tuesday would be Zone One appointment patient...Zone Two will help out to see the outpatient, and then we switch (the schedule)"* [Clinic 3, Professional, 37 years old].

**Care coordinator.**    To cope with the limited space of referral notes in the NCD care form, clinic staffs took the initiative by attaching extra paper to the referred patient information together with NCD care form, as noted by a participant: *"She (referral party) will write there (in extra paper attached) and attached with NCD care form. For more information, please refer the sheet at the back (attachment)"* [Clinic 12, Professional, 47 years old]. For the appointment defaulter tracing activity, three clinics created an appointment defaulter book to facilitate the monitoring, as shared, *"we have an appointment system book, we will count patients that have defaulted from the book"* [Clinic 6, Professional, 35 years old]. Meanwhile, clinics which were equipped with an existing partial information technology (IT) system that was originally intended for billing purposes had made modifications to the system in order to meet the EnPHC requirements on visit checklist and appointment defaulter tracing.

**NCD risk management.**    As a countermeasure to the issues of conducting on-site risk score calculations, clinics traced the necessary laboratory investigation results to calculate the risk score before the patient's appointment day. This action was perceived as necessary to

reduce patients' waiting time and to avoid any misunderstanding regarding the use of phones in front of patients.

**Referrals.** A porter system was created in two clinics to improve the referral system. An assistant medical officer or staff nurse was appointed as a porter to bring the NCD care form to the hospitals to get the referral appointment date for the patients and the hospitals' medical officers would review the referral notes first before assigning the appointment date. This process granted an opportunity for discussion between clinic and hospital. This porter system facilitated the monitoring of hospital appointment defaulters and prevented loss of NCD care forms once the form was given to the patient themselves to get the hospital appointment, as currently practiced.

## Perceived benefit

**Triage counters.** Participants informed that the implementation of triage counter facilitated the management of patient traffic within the clinic, as one participant shared; *"After EnPHC, can see the benefit. . . (we) can channel patient (accordingly)"* [Clinic 1, Professional, 43 years old]. They also shared that by having the primary triage counter, assistant medical officers and staff nurses were empowered to observe patients' condition while waiting for their turn to see the doctors, as mentioned by a participant, *"We can observe patients while they're at the waiting area and take the necessary action when needed"*. (Clinic 11, Staff Nurse, 46 years old).

**Care coordinator.** Despite the challenges, the clinic staffs realised that the NCD care form provides more structured and reader-friendly information for patient care management. Suggestions for a booklet design for the NCD care forms were made to minimise repetitive entry of the demographic information and to minimise the risk of missing documents as the form was merely two sheets of carbonless copy paper. The visit checklist was found not only useful for patients' record management, but it was also a valuable tool to trace appointment defaulters. Care coordinators could detect the appointment defaulters from the list and contact the defaulter to investigate the reason for defaulting and providing new appointment date, as informed by one participant: *"we can trace patient by appointment, filter (patient) by date"* [Clinic 7, Professional, 44 years old]. Performing the appointment defaulter tracing activity was informed to have reduced the number of defaulted appointments. The activity itself has resulted in a closer interpersonal relationship between clinic staff and patients when the clinic staff contacted them. This opinion was shared by participant: *"Patient felt they were being cared more (when) we called them even on Saturday (and) Sunday"* [Clinic 8, Professional, 34 years old].

**NCD screening.** The NCD screening activity was shared to have a positive impact on the number of newly detected NCD cases which could reduce the number of undiagnosed and untreated patients within the community. One noted, *"Our new (NCD) cases tripled since Enhance (EnPHC). Currently, diabetes 20 cases (detected), hypertension (case detected) increase three times. . . after July increase 20 patients (detected) in a month, just hyperlipidaemia."* [Clinic 5, Professional, 47 years old].

**NCD risk management.** The clinics shared that the risk stratification criteria were used as a health education guide by clinic staff for general health, exercise, foot care and smoking habits. In addition, this activity empowered the assistant medical officer and staff nurse to deliver health education and it was no longer restricted by doctors only. With the EnPHC initiative, the District health offices were said to have conducted training sessions with the assistant medical officer and staff nurse to improve their knowledge, skills and experience in delivering health education.

**Referrals.** Enhanced communication was a by-product of this improved referral system. It was found that CC in the clinics and liaison officers from the hospitals had been in close

contact with each other in monitoring patient's adherence to hospital referral appointments, particularly when a patient was reported to have defaulted referral appointment and a new appointment had to be arranged. The communication was performed via various means such as phone call, emails and mobile communication applications.

## Discussion

We adopted a mixed-methods approach to assess the implementation of EnPHC intervention packages. Before an intervention is scaled up, fidelity assessment is crucial to address implementation issues before outcome evaluation to ensure the intervention is delivered as intended while feasibility assessment aims to uncover the strengths, challenges and weaknesses. These assessments are important as the context-specific adaptations can be made for improvement. After two months of the implementation of the EnPHC initiative, the interventions were implemented in all participating clinics. Fidelity towards the implementation is high as the interventions mostly enhance the existing activities. However, the extent of implementation for each intervention component and the execution process differs across clinics. This variation is mainly due to the adaptations made to overcome the clinic-specific shortcomings namely, physical structure, human resource and technology uptake.

While other studies have shown the importance of three criteria in setting up a triage system namely, tolerable health care resource, brief assessment of medical need by health care provider in charge at the triaging point and an established triage plan or system [11], this study reveals an additional criterion, physical location. Pre-assessment on physical location can help to predict the feasibility of its implementation, especially for clinics with old premises [33]. This study discovered that almost all the participating clinics were old but the pre-implementation fidelity assessment for physical location was partially conducted during its selection [24]. By doing the assessment, pre-emptive renovations and refurbishments can be done before the initiative is implemented [33].

The challenge of human resource was prominent in all intervention components, thus appearing as an important factor for the EnPHC initiative's sustainability and scalability. Our findings reveal that the interventions introduce changes in the clinics' work procedures and behaviour but with minimal changes in the staffing and on how the tasks are done—most of the interventions were executed manually with minimal technology assistance. If the issue of human resource availability is likely to persist, there should be considerations on making best use of available resources [34,35]. Alternatively, the world is now in the Industrial Revolution 4.0 era where the central focus is on technology and task-shifting from human-dependent to technology-enhanced innovations; this includes the health sector [36].

Although the health sector strives to make use of the latest technology, skilled HCPs in clinical care are still required [34] for care management. Hence, continuous training sessions to refresh and enhance the HCPs skills on care delivery, health education as well as communication skills are needed to ensure appropriate delivery of services [34,37,38]. For instance, communication for inter-facility case referrals was enhanced by utilising technology (email and mobile communication applications) instead of sticking with the traditional paper-based methods. Good team dynamics, such as teamwork and task sharing among the competent HCPs, can give flexibility to overcome the human resource challenges, without compromising process quality and patient safety [34]. This is in line with the teamwork and task-shifting of NCD risk management observed in the clinics. The process quality and appropriateness of care delivered through these interventions will have to be addressed should the EnPHC initiative is to be scaled-up.

Technology uptake such as the electronic medical record can reduce the corporeal efforts related to care provision and service quality [36,39]. In our study, this approach can be applied to the visit checklist and clinic appointment scheduling, NCD care form management of records, the NCD risk score calculations as well as arrangement of referrals between clinic and hospital. Feasibility of technology implementation in Malaysia, however, should also consider the existing infrastructure of the building where it will be implemented, specifically on the electrical wiring, network access, electronic equipment required as well as users' technology literacy to use any proposed technology to be implemented [36,40].

The clinics setting is an influencing factor for the adaptive measures taken which has been noted as a domain for implementation [35]. An open space physical counter setup can create transparency and help to build awareness and understanding among patients regarding the activities performed by HCPs, as transparent accountability can prevent harm [41]. However, the acceptance of open concepts at the clinic varied. It was contrary to the privacy concerns raised by the patients to the HCPs in eleven of the participating clinics, which resulted in dedicated rooms or setting up privacy screens for secondary triaging. Patient norms and behaviour is another component which has to be considered when introducing any new intervention at the primary healthcare setting. The use of mobile applications for NCD risk management created negative patient perception on its use which needs to be countered. Despite the reminders given, many appointment defaulters continued defaulting due to the patients' dependency on family members to bring them to the clinics. This highlighted the importance to explore the service-users' perspectives on the planned interventions [42].

Extensive follow-up actions by the HCPs when a patient was unreachable by phone were a testament to the high value placed on defaulter tracing. Studies have also reported defaulter tracing as a favourable method of community engagement and to foster HCP-patient relationship [43]; this has been borne out in our study whereby the HCPs perceive that defaulter tracing activity have fostered a closer interpersonal relationship between clinic staff and patients. The manner and appropriateness of how the communication is conveyed also influences patients' trust which then appears as a key factor for the care-seeking behaviour [44] and the person-centred care concept [45]. This adaptation to the appointment defaulter tracing activity should be considered for future scalability.

The person-centred care concept also stresses the importance of a patient's care continuity across facilities [45]. Thus, the communication established between clinics and hospitals on referral management is significant. The established communication, however, is a foundation of future collaboration efforts between the facilities. Participating EnPHC clinics should also share with each other their experiences and the local adaptations in overcoming challenges in their respective clinics to create a platform of discussion and a knowledge pool in seeking solutions [46].

## Strengths & limitations

This study's strength lies on its data triangulation efforts. Its mixed methods data collection approach enabled multiple perspectives of the subject being evaluated. The involvement of multidisciplinary research team members [9], scholarly interaction and sharing of ideas were encouraged, resulting in a triangulation of views and opinions before reaching consensus. It provides a richer and more comprehensive understanding of the implementation fidelity and feasibility issues encountered by HCPs.

This study also had its limitations on the data collection process. The structured observation using checklist was conducted in cross-sectional manner which only captured that particular period of observation and might not reflect the whole process. However, the subsequent IDI

helped to fill the information gaps. Even though the IDI participants were purposively selected as the representative with the best knowledge on the implementation process in their clinics, 'the person with best knowledge' was perceived from each clinic's consensus; there were no quantifiable evaluation assessment made to measure the participant's actual knowledge. Additionally, it might also have limited the sharing of challenges encountered by other sections within the clinics. The scope of evaluation was limited to the organisational (or structural) process; the clinical processes were excluded as it was evaluated by another evaluation team. Data collection at two months after the interventions were rolled out was another limitation as some clinics were still in the adjustment period. It led to the research team's decision to conduct a follow-up evaluation study planned in the tenth month of implementation to explore the implementation feasibility across time in the participating clinics. Focus group discussions with other personnel in the clinic will also be conducted to obtain a more representative feedback from the clinics.

## Conclusion

The EnPHC initiative is a model of service delivery based on comprehensive patient-centred care particularly on the PHC perspective in managing NCD. The evaluation study has disclosed issues of human resource and physical infrastructure when a supplementary intervention is implemented. It highlighted the need for a good organisational process to address these issues which are equally important as having good clinical practices and processes. The adaptive measures taken to suit each clinic's local contexts are also important considerations for the program owners to ensure the initiative's success and sustainability.

## Supporting information

**S1 Checklist.**
(PDF)

## Acknowledgments

Declarations

The authors would like to declare our gratitude to the Director General of Health Malaysia for his permission to publish this paper. A special thanks to the State Directors of Johor and Selangor and their staffs who facilitated the conduct of this research, and the Harvard T. H. Chan School of Public Health for their inputs on the study protocols. We would like to extend our thanks to members of the EnPHC-PE research team. The authors are also immensely grateful to all participants who participated in this study. Last but not least, we thank Associate Professor. Dr. Kamaliah Mohamad Noh, from University of Cyberjaya Malaysia, who had helped in editing and proof read the manuscript.

## Author Contributions

**Conceptualization:** Lee Lan Low, Fathullah Iqbal A. B. Rahim.

**Data curation:** Lee Lan Low, Fathullah Iqbal A. B. Rahim.

**Formal analysis:** Lee Lan Low, Fathullah Iqbal A. B. Rahim.

**Investigation:** Lee Lan Low, Fathullah Iqbal A. B. Rahim.

**Methodology:** Lee Lan Low, Fathullah Iqbal A. B. Rahim.

**Project administration:** Lee Lan Low.

**Resources:** Lee Lan Low.

**Validation:** Lee Lan Low.

**Writing – original draft:** Lee Lan Low, Fathullah Iqbal A. B. Rahim, Nur Aqlili Riana Hamzah, Mohd Safiee Ismail.

**Writing – review & editing:** Lee Lan Low, Fathullah Iqbal A. B. Rahim, Nur Aqlili Riana Hamzah, Mohd Safiee Ismail.

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
