## [Decision Letter · Decision Letter 0]

16 Jul 2020

PONE-D-20-04093

Process evaluation of enhancing primary health care for non-communicable disease management: uncovered the fidelity & feasibility elements

PLOS ONE

Dear Dr. LOW,

Thank you for submitting your manuscript to PLOS ONE. After careful consideration, we feel that it has merit but does not fully meet PLOS ONE’s publication criteria as it currently stands. Therefore, we invite you to submit a revised version of the manuscript that addresses the points raised during the review process.

Please see the comments below.

We look forward to receiving your revised manuscript.

Kind regards,

Fernando C. Wehrmeister

Academic Editor

PLOS ONE

Additional Editor Comments:

The manuscript addressed an important research question, especially in a middle income setting. As pointed out by the reviewers, the methods section needs to be improved for a better understanding of the study. Also, improvements in discussion is needed.

2. Please provide additional details regarding participant consent.

In the ethics statement in the Methods and online submission information, please ensure that you have specified (i) whether consent was informed and (ii) what type you obtained (for instance, written or verbal, and if verbal, how it was documented and witnessed).

If your study included minors, state whether you obtained consent from parents or guardians.

If the need for consent was waived by the ethics committee, please include this information.”

3. Please include additional information regarding the survey or questionnaire used in the study and ensure that you have provided sufficient details that others could replicate the analyses.

For instance, if you developed a questionnaire as part of this study and it is not under a copyright more restrictive than CC-BY, please include a copy, in both the original language and English, as Supporting Information.

6. Please include a separate caption for each figure in your manuscript.

7. Your ethics statement must appear in the Methods section of your manuscript. If your ethics statement is written in any section besides the Methods, please move it to the Methods section and delete it from any other section. Please also ensure that your ethics statement is included in your manuscript, as the ethics section of your online submission will not be published alongside your manuscript.

Reviewers' comments:

Reviewer's Responses to Questions

**Comments to the Author**

1. Is the manuscript technically sound, and do the data support the conclusions?

Reviewer #1: Partly

Reviewer #2: Partly

2. Has the statistical analysis been performed appropriately and rigorously? 

Reviewer #1: Yes

Reviewer #2: N/A

3. Have the authors made all data underlying the findings in their manuscript fully available?

Reviewer #1: Yes

Reviewer #2: No

4. Is the manuscript presented in an intelligible fashion and written in standard English?

Reviewer #1: No

Reviewer #2: Yes

5. Review Comments to the Author

Reviewer #1: The manuscript attempts to address an important question by evaluating a primary healthcare initiative to improve NCD care management in Malaysia. The findings are rich and informative, but need to be presented in a more intelligible fashion. The manuscript as it currently stands has potential but lacks rigor and detail, and needs major revisions.

Specifically:

1. It is unclear how utilization was defined and assessed.

2. How were the 20 public primary health care centers sampled/identified? The authors do not describe this.

3. The authors mention that data collection was conducted 2 months vs 3-5 months after the program was implemented, but do not mention how these two different data collection time points may have impacted the results. Given that 2 months of implementation is very short, implementation differences may have been observed between clinics that had data collection at 2 months vs 5 months?

4. Were the IDIs unstructured? semi-structured? This should be described in the methods. Additionally, authors should include the data collection instrument as an appendix to the manuscript if possible.

5. Authors do not explain the rationale for selecting only 1 participant per clinic. Additionally, the participants varied in terms of their role/responsibility in the clinic, therefore the breadth of implementation components evaluated (fidelity, feasibility, adaptation, benefit) could have been different from one clinic to another depending on the participant's level of exposure/role. This should be mentioned as a potential limitation, given there was only 1 participant per clinic.

6. There is insufficient detail to describe how much the checklist changed between the two iterations.

7. Figure 1 is not very informative and does not add to the manuscript. Its findings should be covered in the results section text.

8. The results section is somewhat difficult to follow, and authors have combined background/context elements, evidence from other studies (which needs to be moved to the discussion section) and results from this particular study. The results section needs restructuring. It should also clearly summarize the key implementation components being evaluated (fidelity, feasibility, adaptation, benefit).

9. It would be informative to tell the reader how much is "some clinics", "few clinics", in the results. Perhaps add a proportion, so that the reader can assess how common some challenges were across the 20 clinics.

10. The discussion section needs to be strengthened to clearly summarize the key findings in terms of fidelity, feasibility, adaptation and benefit of the program being evaluated, and to put them into context with other studies/programs, etc.

11. Authors should also state key recommendations based on their findings.

12. Careful grammar editing is needed throughout the manuscript.

Reviewer #2: NCDs are relevant and increasing public health issues in low- and middle-income countries, but often neglected as priority for health interventions or health research issues. The paper is interesting and reads well. However, please see below a few comments aiming to improve it.

Title:

Geographic area to include in the title (e.g. “…..in two states in Malaysia”).

Background:

Lines 49 to 61 in the background seem more appropriate as methods points, and preferable to move to the methods section, namely the study design and sampling.

Methods:

Study design and sampling

The description of the study design needs to be developed. You can move in that section a few methods aspects developed in the background as per comment above.

The sampling section seems a bit weak and needs to be deeply improved. It was indicated that the intervention was implemented in 40 public health care clinics (20 control clinics and 20 intervention clinics) at two selected states. Do the 20 PHC of the evaluation only include the 20 intervention clinics or both intervention and control clinics? In the event, both type of clinics are included, can you explicit the process for sample size calculation and whether you assessed the precision and power of the sample to be able to detect significant difference between intervention and control clinics? In the event the 20 clinics for the evaluation are intervention clinics only, can you explain the rationale of focusing on the intervention clinics only as you may have missed a good opportunity to shape an interesting and relevant design comparing intervention and control clinics? Additionally whether you focus on the intervention clinics, can you provide an explanation with respect to the choice of the two states for the intervention and try to discuss how they may (or not) be representative of the overall national situation as the assessment aims to end up to evidences that may help for scaling up the intervention?

Results:

The quality of figures on the PDF format seems poor and needs to be improved. Moreover, the figures need to be numbered properly and well referenced in the text, and figures titles need to be clear and more explicit with respect to the information presented.

For statements like “The current availability of human resource for health in Malaysia is still low when compared to OECD countries” (page 10), it would be good to provide ratios or estimates whether available.

Whether feasible and data available, it would be desirable to have more results. For instance, it may be interesting to know to which extent the secondary triage may have helped improving the detection of undiagnosed NCD cases for ensuring care continuity (lines 186-187). From the PHC register, are there collected data with respect to the patients that may be used for diagnosis and patient case classification? In the same vein, when you say that longer patient waiting time was observed in clinics with insufficient personnel to perform tasks at secondary triage counter (lines 190-191), it will be worthwhile trying to quantity the ‘longer patient waiting time’; or quantitative information in tracing appointment defaulter (line 234) and patients’ adherence to appointments (line 250); or estimating NCD screening activities performed properly according to main reference document for NCD screening (line 259; 265, 266, 270, 271)), or to which extent the referral system was improved (line 311), etc.

Discussions:

It would be desirable to shape the discussions section as per the evaluation-focused four dimensions i.e. fidelity, feasibility, adaptation and benefit.

Concerning the design and dimensions of the evaluation, it makes sense and it is relevant as you assessed the ‘structural quality’ of the intervention (staff, infrastructure, structure, organization, etc.), but it may have also been worthwhile trying to address the process quality for such an intervention (e.g. appropriate triage/assessment of NCDs patients health, appropriate examination of patients, and appropriate diagnosis, or appropriate treatment, referral and counseling, etc.) as that would have been relevant for such an evaluation with the aim to scale-up the intervention package. You may have made the choice to focus on the ‘structural quality’, but good to justify more that choice in the discussions as well as the choice why the design did not address the process quality of care.

It may also be relevant to discuss the scale-up opportunity and challenges with respect to scaling up such interventions in other regions or nationwide.

Format:

Review the text and double-check typo issues (e.g. line 74 “being introduce”, line 234 “Some clinic has initiated by creating their own”, etc.)

6. PLOS authors have the option to publish the peer review history of their article (what does this mean?). If published, this will include your full peer review and any attached files.

Reviewer #1: No

---

## [Author Response · Author response to Decision Letter 0]

15 Oct 2020

Editor Comments

Additional Editor Comments:

The manuscript addressed an important research question, especially in a middle income setting. As pointed out by the reviewers, the methods section needs to be improved for a better understanding of the study. Also, improvements in discussion is needed.

Thank you, noted & had addressed it

Thank you, noted & had addressed it

2. Please provide additional details regarding participant consent.

In the ethics statement in the Methods and online submission information, please ensure that you have specified (i) whether consent was informed and (ii) what type you obtained (for instance, written or verbal, and if verbal, how it was documented and witnessed).

If your study included minors, state whether you obtained consent from parents or guardians.

If the need for consent was waived by the ethics committee, please include this information.”

Thank you, noted & had moved the ethics statement to methods section

Ref: Pg.11, line 140-147

3. Please include additional information regarding the survey or questionnaire used in the study and ensure that you have provided sufficient details that others could replicate the analyses.

Thank you, noted & had added as [Appendix]

Ref: Pg.9, line 95-96

We have included the following information in the manuscript:

The dataset which support the findings of this article belongs to the Enhanced Primary Healthcare Evaluation study. Request for data can be obtained from Dr. Mohd Azahadi Omar (drazahadi@moh.gov.my), the head of sector for Biostatistics & Data Repository, National Institute of Health, Ministry of Health Malaysia and with the permission from the Director General of Health, Malaysia.

Ref: Pg.24, line 437-442

5. PLOS requires an ORCID iD for the corresponding author in Editorial Manager on papers submitted after December 6th, 2016. Please ensure that you have an ORCID iD and that it is validated in Editorial Manager.

Thank you, noted & had addressed it

The corresponding author’s ORCID 0000-0002-9952-3017 

6. Please include a separate caption for each figure in your manuscript.

Thank you, noted & had addressed it

7. Your ethics statement must appear in the Methods section of your manuscript. If your ethics statement is written in any section besides the Methods, please move it to the Methods section and delete it from any other section. Please also ensure that your ethics statement is included in your manuscript, as the ethics section of your online submission will not be published alongside your manuscript.

Thank you, noted & had moved to methods section

Ref: Pg.11, line 140-147

 

REVIEWER #1 

Reviewer #1: The manuscript attempts to address an important question by evaluating a primary healthcare initiative to improve NCD care management in Malaysia. The findings are rich and informative, but need to be presented in a more intelligible fashion. The manuscript as it currently stands has potential but lacks rigor and detail, and needs major revisions.

Thank you and had responded to each point 

1.It is unclear how utilization was defined and assessed.

Thank you for highlighted this. We have added this statement.

Ref: Pg.8, line 61-62

Fidelity was assessed from two perspectives; availability of the implemented intervention and its utilisation according to guidelines and protocols. 

(notes: Assessing the utilisation is to ensure the availability of some of intervention components such as triage counter need to have a person to handle the counter, similarly with the NCD care form and screening form, are being used by the clinics.) 

2. How were the 20 public primary health care centers sampled/identified? The authors do not describe this.

Thank you for highlighted this. We have added this statement.

Ref: Pg. 6-7, Line 30-39

The initiative was piloted in July 2017 in 20 public primary care clinics in two states of Johor and Selangor in Malaysia. Selection of the two states was based on the balance between regional representativeness, intervention budget and implementation capacity which involves manpower and facilities [22]. For the purpose of evaluation, an addition of 20 control clinics were selected, matched according to urban-rural stratification, clinic setting and type as well as human resource availability in the clinic. Three evaluation studies on the EnPHC initiative were conducted concurrently and independently; two outcome evaluation at the community and facility level [23, 24], and process evaluation focusing on the implementation process [9]. Since our focus was on the implementation fidelity and feasibility, the control clinics were excluded from the process evaluation. 

3. The authors mention that data collection was conducted 2 months vs 3-5 months after the program was implemented, but do not mention how these two different data collection time points may have impacted the results. Given that 2 months of implementation is very short, implementation differences may have been observed between clinics that had data collection at 2 months vs 5 months?

Thank you. It was an error on our side and we have made the correction. 

Ref: Pg.8, line 71-76

The mixed method study design using structured observation and in-depth interview (IDI) was conducted from September to November 2017, two months after EnPHC was rolled out in the 20 participating public primary health care (PHC) clinics. As the clinic setting, infrastructure and human resource varied across the clinics, the adjustments in implementation were expected to be different. Hence, two months were deemed appropriate and sufficient period for the clinics for adjustments.

4. Were the IDIs unstructured? semi-structured? This should be described in the methods. Additionally, authors should include the data collection instrument as an appendix to the manuscript if possible.

Thank you and we have revised. We have included the data collection instrument as an appendix for this manuscript.

Ref: Pg.9-Page 10, Line 105-109

The in-depth interview (IDI) – semi-structured interviews were conducted with 20 HCPs respectively from each intervention clinic. Selection of interviewees was purposive to reflect sharing of implementation issues across clinic settings by the different categories of staff holding different roles and responsibilities. IDI aimed to explore the implementation issues and to clarify information gaps from the self-assessment form and/or the observation. 

Ref: Pg.9, line 95-96

Refer Appendix for the structured observation checklist

5. Authors do not explain the rationale for selecting only 1 participant per clinic. Additionally, the participants varied in terms of their role/responsibility in the clinic, therefore the breadth of implementation components evaluated (fidelity, feasibility, adaptation, benefit) could have been different from one clinic to another depending on the participant's level of exposure/role. This should be mentioned as a potential limitation, given there was only 1 participant per clinic.

Thank you. We have addressed the selection in Methods, and also included its limitation under Strengths and limitations.

Ref: Pg.9, line 95-96

Selection of interviewees was purposive to reflect sharing of implementation issues across clinic settings by the different categories of staff holding different roles and responsibilities.

 Ref: Pg.22, line 403-407

Even though the IDI participants were purposively selected as the representative with the best knowledge on the implementation process in their clinics, ‘the person with best knowledge’ was perceived from each clinic’s consensus; there were no quantifiable evaluation assessment made to measure the participant’s actual knowledge.

6. There is insufficient detail to describe how much the checklist changed between the two iterations.

Thank you for this question.

Ref: Pg.9, line 92-95

Information during pre-testing was captured as field notes which was used to improve the checklist, specifically on the observations for the ISS intervention and document formatting to create more space for researchers to add notes. 

7. Figure 1 is not very informative and does not add to the manuscript. Its findings should be covered in the results section text.

Thank you for the suggestion. We have replaced this Figure with the following text. 

Ref: Pg.12-13, line 158-181

Fidelity assessment was based on the availability of the intervention and its utilisation in accordance to guideline protocols. In general, all intervention components have been implemented according to the protocols in all 20 clinics, except for the following specific components: 

Triage Counters (primary triage counter). All clinics deployed the primary triage counter except for one clinic which failed due to limitation in building infrastructure. Four primary triage counters from the remaining 19 clinics were found unattended and acted as placeholders. The secondary triage counter, however, was successfully set up and utilised in all clinics.

Triage Counters (primary triage counter). All clinics deployed the primary triage counter except for one clinic which failed due to limitation in building infrastructure. The secondary triage counter, however, was successfully set up in all clinics.

Care Coordinator (visit checklist). One clinic failed to utilise the required format of the visit checklist as they deemed the existing patient appointment card sufficient to capture the required information.

Care Coordinator (appointment defaulter tracing). The appointment defaulter tracing activity was successfully conducted in all clinics. However, as the there was no guideline protocol on how it should be conducted, fidelity assessment could not be made as there was no standard to match. This issue was cited to contribute into feasibility issues during further exploration through IDI. 

Referrals (external referral). The communication between clinics and hospitals on referral management had been established in all clinics. However, fidelity assessment was unsuccessfully conducted on the communication process as the guideline has no specific criteria to be communicated in the referral as standards to be met.

8. The results section is somewhat difficult to follow, and authors have combined background/context elements, evidence from other studies (which needs to be moved to the discussion section) and results from this particular study. The results section needs restructuring. It should also clearly summarize the key implementation components being evaluated (fidelity, feasibility, adaptation, benefit).

Thank you for the suggestion. We have restructured the entire Results and Discussion section.

(Results) Ref: Pg.11-19, line 149-313

(Discussion) Ref: Pg.19-23, line 314-412

9. It would be informative to tell the reader how much is "some clinics", "few clinics", in the results. Perhaps add a proportion, so that the reader can assess how common some challenges were across the 20 clinics.

Thank you, we have quantified these words, where possible. 

Ref: Pg.14, line 210

Eight of the clinics even took the extra initiative of doing home visits, but it brought in logistical challenges of tracing houses when it came to incomplete addresses. 

Ref: Pg.15, line 234

Triage Counters. To overcome the problem, modifications of the primary triage counter were carried out in four clinics were carried out to also function as a registration counter for new patient visits.

Ref: Pg.16, line 256

For the appointment defaulter tracing activity, three clinics created an appointment defaulter book to facilitate the monitoring,

Ref: Pg.16, line 237

The rooms were viewed as necessary in eleven clinics for privacy reasons… 

10. The discussion section needs to be strengthened to clearly summarize the key findings in terms of fidelity, feasibility, adaptation and benefit of the program being evaluated, and to put them into context with other studies/programs, etc.

Thank you for suggestion. We have restructured the Discussion section.

Ref: Pg.19-23, line 314-412

11. Authors should also state key recommendations based on their findings.

Thank you for suggestion. The recommendations were embedded in each of discussion point. 

Ref: Pg.20, line 339-341

If the issue of human resource availability is likely to persist, there should be considerations on making best use of available resources [34, 35]. 

Ref: Pg.21, line 360-363

Feasibility of technology implementation in Malaysia, however, should also consider the existing infrastructure of the building where it will be implemented, specifically on the electrical wiring, network access, electronic equipment required as well as users’ technology literacy to use any proposed technology to be implemented.

Ref: Pg.21, line 370-376

Patient norms and behaviour is another component which has to be considered when introducing any new intervention at the primary healthcare setting…This highlighted the importance to explore the service-users’ perspectives on the planned interventions.

Ref: Pg.21-22, line 378-385

Studies have also reported defaulter tracing as a favourable method of community engagement and to foster HCP-patient relationship [43]; this has been borne out in our study whereby the HCPs perceive that defaulter tracing activity have fostered a closer interpersonal relationship between clinic staff and patients.

Ref: Pg.22, line 386-392

The person-centred care concept also stresses the importance of a patient’s care continuity across facilities [45]. Thus, the communication established between clinics and hospitals on referral management is significant. The established communication, however, is a foundation of future collaboration efforts between the facilities. Participating EnPHC clinics should also share with each other their experiences and the local adaptations in overcoming challenges in their respective clinics to create a platform of discussion and a knowledge pool in seeking solutions.

12. Careful grammar editing is needed throughout the manuscript.

Thank you. We have sent the manuscript for editing and proof reading.

REVIEWER #2

NCDs are relevant and increasing public health issues in low- and middle-income countries, but often neglected as priority for health interventions or health research issues. The paper is interesting and reads well. However, please see below a few comments aiming to improve it.

Thank you and had responded to each point

Title:

Geographic area to include in the title (e.g. “…..in two states in Malaysia”).

Thank you for suggestion. We have included Malaysia.

Process evaluation of enhancing primary health care for non-communicable disease management in Malaysia : uncovering the fidelity & feasibility elements 

We chose not to include the “two states” as consideration for selection has been made accordingly by the program owners. We have also included this information in the manuscript (Ref: Pg. 6, Line 31-33).

Background:

Lines 49 to 61 in the background seem more appropriate as methods points, and preferable to move to the methods section, namely the study design and sampling.

Thank you for suggestion. We have moved to methods

Ref: Pg.7-8, line 56-69

Methods:

Study design and sampling

The description of the study design needs to be developed. You can move in that section a few methods aspects developed in the background as per comment above.

Thank you for suggestion. We have created Box 1 to explain the intervention components in details. 

Ref: Pg.5-6, line 28, Box 1: The EnPHC Intervention Components

The sampling section seems a bit weak and needs to be deeply improved. It was indicated that the intervention was implemented in 40 public health care clinics (20 control clinics and 20 intervention clinics) at two selected states. Do the 20 PHC of the evaluation only include the 20 intervention clinics or both intervention and control clinics? In the event, both type of clinics are included, can you explicit the process for sample size calculation and whether you assessed the precision and power of the sample to be able to detect significant difference between intervention and control clinics? In the event the 20 clinics for the evaluation are intervention clinics only, can you explain the rationale of focusing on the intervention clinics only as you may have missed a good opportunity to shape an interesting and relevant design comparing intervention and control clinics? Additionally whether you focus on the intervention clinics, can you provide an explanation with respect to the choice of the two states for the intervention and try to discuss how they may (or not) be representative of the overall national situation as the assessment aims to end up to evidences that may help for scaling up the intervention?

Thank you for highlighted this. We have made the necessary corrections to convey a clearer meaning regarding the sampling.

Ref: Pg. 6-7, Line 30-39

The initiative was piloted in July 2017 in 20 public primary care clinics in two states of Johor and Selangor in Malaysia. Selection of the two states was based on the balance between regional representativeness, intervention budget and implementation capacity which involves manpower and facilities [22]. For the purpose of evaluation, an addition of 20 control clinics were selected, matched according to urban-rural stratification, clinic setting and type as well as human resource availability in the clinic. Three evaluation studies on the EnPHC initiative were conducted concurrently and independently; two outcome evaluation at the community and facility level [23, 24], and process evaluation focusing on the implementation process [9]. Since our focus was on the implementation fidelity and feasibility, the control clinics were excluded from the process evaluation. 

Results:

The quality of figures on the PDF format seems poor and needs to be improved. Moreover, the figures need to be numbered properly and well referenced in the text, and figures titles need to be clear and more explicit with respect to the information presented.

Thank you for suggestion. As this comment was in line with Reviewer #1’s comments, we have decided to drop this Figure and replace it in text. 

Ref: Pg.12-13, line 158-181

Fidelity assessment was based on the availability of the intervention and its utilisation in accordance to guideline protocols. In general, all intervention components have been implemented according to the protocols in all 20 clinics, except for the following specific components: 

Triage Counters (primary triage counter). All clinics deployed the primary triage counter except for one clinic which failed due to limitation in building infrastructure. Four primary triage counters from the remaining 19 clinics were found unattended and acted as placeholders. The secondary triage counter, however, was successfully set up and utilised in all clinics.

Triage Counters (primary triage counter). All clinics deployed the primary triage counter except for one clinic which failed due to limitation in building infrastructure. The secondary triage counter, however, was successfully set up in all clinics.

Care Coordinator (visit checklist). One clinic failed to utilise the required format of the visit checklist as they deemed the existing patient appointment card sufficient to capture the required information.

Care Coordinator (appointment defaulter tracing). The appointment defaulter tracing activity was successfully conducted in all clinics. However, as the there was no guideline protocol on how it should be conducted, fidelity assessment could not be made as there was no standard to match. This issue was cited to contribute into feasibility issues during further exploration through IDI. 

Referrals (external referral). The communication between clinics and hospitals on referral management had been established in all clinics. However, fidelity assessment was unsuccessfully conducted on the communication process as the guideline has no specific criteria to be communicated in the referral as standards to be met.

For statements like “The current availability of human resource for health in Malaysia is still low when compared to OECD countries” (page 10), it would be good to provide ratios or estimates whether available.

Whether feasible and data available, it would be desirable to have more results. For instance, it may be interesting to know to which extent the secondary triage may have helped improving the detection of undiagnosed NCD cases for ensuring care continuity (lines 186-187). From the PHC register, are there collected data with respect to the patients that may be used for diagnosis and patient case classification? In the same vein, when you say that longer patient waiting time was observed in clinics with insufficient personnel to perform tasks at secondary triage counter (lines 190-191), it will be worthwhile trying to quantity the ‘longer patient waiting time’; or quantitative information in tracing appointment defaulter (line 234) and patients’ adherence to appointments (line 250); or estimating NCD screening activities performed properly according to main reference document for NCD screening (line 259; 265, 266, 270, 271)), or to which extent the referral system was improved (line 311), etc.

Thank you for suggestion. As this comment was in line with Reviewer #1’s comments, we have structured the entire Results section according to the four aspects of fidelity, feasibility, adaptation and benefit. However, we are unable to provide the quantifiable information on long waiting time and patient data since this study focused on the structural process of the implementation instead of the clinical processes.

Ref: Pg.11-19, line 149-313

Discussions:

It would be desirable to shape the discussions section as per the evaluation-focused four dimensions i.e. fidelity, feasibility, adaptation and benefit.

Thank you for suggestion. As this comment was in line with Reviewer #1’s comments, we have structured the entire Discussion section.

Ref: Pg.19-23, line 314-412

Concerning the design and dimensions of the evaluation, it makes sense and it is relevant as you assessed the ‘structural quality’ of the intervention (staff, infrastructure, structure, organization, etc.), but it may have also been worthwhile trying to address the process quality for such an intervention (e.g. appropriate triage/assessment of NCDs patients health, appropriate examination of patients, and appropriate diagnosis, or appropriate treatment, referral and counseling, etc.) as that would have been relevant for such an evaluation with the aim to scale-up the intervention package. You may have made the choice to focus on the ‘structural quality’, but good to justify more that choice in the discussions as well as the choice why the design did not address the process quality of care.

Thank you. We have included your concern as our study limitation.

Ref: Pg.23, line 407-408

The scope of evaluation was limited to the organisational (or structural) process; the clinical processes were excluded as it was evaluated by another evaluation team.

It may also be relevant to discuss the scale-up opportunity and challenges with respect to scaling up such interventions in other regions or nationwide.

Thank you. After the restructuring of the Discussion section, we have included your input as embedded recommendations within the Discussion. Some examples of the embedded statements are:

Ref: Pg.19-20, line 329-330

Pre-assessment on physical location can help to predict the feasibility of its implementation, especially for clinics with old premises [33].

Ref: Pg.20, line 332-334

By doing the assessment, pre-emptive renovations and refurbishments can be done before the initiative is implemented [33].

Ref: Pg.20, line 339-341

If the issue of human resource availability is likely to persist, there should be considerations on making best use of available resources [34, 35].

Ref: Pg.20-21, line 353-355

The process quality and appropriateness of care delivered through these interventions will have to be addressed should the EnPHC initiative is to be scaled-up.

Ref: Pg.21, line 356-357

Technology uptake such as the electronic medical record can reduce the corporeal efforts related to care provision and service quality [36, 39]

Ref: Pg.21, line 360-363

Feasibility of technology implementation in Malaysia, however, should also consider the existing infrastructure of the building where it will be implemented, specifically on the electrical wiring, network access, electronic equipment required as well as users’ technology literacy to use any proposed technology to be implemented [36, 40].

Ref: Pg.21, line 370-376

Patient norms and behaviour is another component which has to be considered when introducing any new intervention at the primary healthcare setting…This highlighted the importance to explore the service-users’ perspectives on the planned interventions [42].

Ref: Pg.22, line 381-385

The manner and appropriateness of how the communication is conveyed also influences patients’ trust which then appears as a key factor for the care-seeking behaviour [44] and the person-centred care concept [45]. This adaptation to the appointment defaulter tracing activity should be considered for future scalability.

Format:

Review the text and double-check typo issues (e.g. line 74 “being introduce”, line 234 “Some clinic has initiated by creating their own”, etc.)

Thank you. We have sent the manuscript for editing and proof reading.

---

## [Decision Letter · Decision Letter 1]

20 Nov 2020

PONE-D-20-04093R1

Process evaluation of enhancing primary health care for non-communicable disease management in Malaysia: uncovering the fidelity & feasibility elements

PLOS ONE

Dear Dr. LOW,

Thank you for submitting your manuscript to PLOS ONE. After careful consideration, we feel that it has merit but does not fully meet PLOS ONE’s publication criteria as it currently stands. Therefore, we invite you to submit a revised version of the manuscript that addresses the points raised during the review process.

We look forward to receiving your revised manuscript.

Kind regards,

Fernando C. Wehrmeister

Academic Editor

PLOS ONE

Additional Editor Comments (if provided):

The manuscript had improved. However, minor details should be addressed. Please see the reviewer comments.

Reviewers' comments:

Reviewer's Responses to Questions

**Comments to the Author**

1. If the authors have adequately addressed your comments raised in a previous round of review and you feel that this manuscript is now acceptable for publication, you may indicate that here to bypass the “Comments to the Author” section, enter your conflict of interest statement in the “Confidential to Editor” section, and submit your "Accept" recommendation.

Reviewer #2: (No Response)

2. Is the manuscript technically sound, and do the data support the conclusions?

Reviewer #2: Yes

3. Has the statistical analysis been performed appropriately and rigorously? 

Reviewer #2: N/A

4. Have the authors made all data underlying the findings in their manuscript fully available?

Reviewer #2: No

5. Is the manuscript presented in an intelligible fashion and written in standard English?

Reviewer #2: Yes

6. Review Comments to the Author

Reviewer #2: Thanks to the authors for addressing the comments.

It was a good idea to describe the intervention components using a box (Box 1). However, you provided too much detail, so the box is on two pages. Ideally, we expect an embedded box to be one page as an embedded figure. Therefore, it will be nice to cut down a bit and fit on one page.

As per the comments, you tried to describe further the sampling process (Page 6-7, line 30-39). However, the place (Background) you inserted this description does not seem appropriate. In the methods, there is a section titled “Study design and sampling”, but there is no information regarding the sampling in that section. I suggest that you move the description of the sampling process (Page 6-7, line 30-39) from the background to the section “Study design and sampling”.

There was also a comment regarding the sample size. Since you have excluded the control clinic, there was no longer a need to address the comment related to the power of the sample to detect significant difference between the intervention and control clinics. However, the comment remains relevant concerning the precision of the sample. It is evident that a sample of 20 clinics and 20 health workers seems small and would not provide a good sample precision. It may therefore be worth addressing that issue (small sample size) as a limitation of the study.

Concerning the discussions, you made considerable efforts to reshape this section and provided valuable additional information.

Otherwise, the authors have well addressed the remaining comments.

Best wishes.

7. PLOS authors have the option to publish the peer review history of their article (what does this mean?). If published, this will include your full peer review and any attached files.

Reviewer #2: **Yes: **Abdoulaye Maïga, PhD

---

## [Author Response · Author response to Decision Letter 1]

18 Dec 2020

Editor Comments

Additional Editor Comments: The manuscript had improved. However, minor details should be addressed. Please see the reviewer comments

The authors’ response - Thank you, noted and had addressed it

Reviewer #2: 

1. Thanks to the authors for addressing the comments. I t was a good idea to describe the intervention components using a box (Box 1). However, you provided too much detail, so the box is on two pages. Ideally, we expect an embedded box to be one page as an embedded figure. Therefore, it will be nice to cut down a bit and fit on one page.

The authors’ response - Thank you for your suggestion. We have further summarised the contents for Box 1 and also formatted the section with single spacing. (Ref: page 5)

2. As per the comments, you tried to describe further the sampling process (Page 6-7, line 30-39). However, the place (Background) you inserted this description does not seem appropriate. In the methods, there is a section titled “Study design and sampling”, but there is no information regarding the sampling in that section. I suggest that you move the description of the sampling process (Page 6-7, line 30-39) from the background to the section “Study design and sampling”.

The authors’ response - Thank you for highlighting this. We have moved the paragraph on sampling process to the “Study design and sampling” section. (Ref: Pg. 6-7 , line 49 – 58)

3. There was also a comment regarding the sample size. Since you have excluded the control clinic, there was no longer a need to address the comment related to the power of the sample to detect significant difference between the intervention and control clinics. However, the comment remains relevant concerning the precision of the sample. It is evident that a sample of 20 clinics and 20 health workers seems small and would not provide a good sample precision. It may therefore be worth addressing that issue (small sample size) as a limitation of the study.

The authors’ response - Thank you for highlighting this. We have added the following information into the section on limitation. (Ref: Pg. 22 Line 410-411 and 416-418)

---

## [Editor Report · Decision Letter 2]

23 Dec 2020

Process evaluation of enhancing primary health care for non-communicable disease management in Malaysia: uncovering the fidelity & feasibility elements

PONE-D-20-04093R2

Dear Dr. LOW,

We’re pleased to inform you that your manuscript has been judged scientifically suitable for publication and will be formally accepted for publication once it meets all outstanding technical requirements.

Kind regards,

Fernando C. Wehrmeister

Academic Editor

PLOS ONE

Additional Editor Comments (optional):

The paper had improved since its first version. Congratulations to the authors for the excellent job done.
---

## [Editor Report · Acceptance letter]

29 Dec 2020

PONE-D-20-04093R2 

Process evaluation of enhancing primary health care for non-communicable disease management in Malaysia: uncovering the fidelity & feasibility elements 

Dear Dr. LOW:

I'm pleased to inform you that your manuscript has been deemed suitable for publication in PLOS ONE. Congratulations! Your manuscript is now with our production department. 

Kind regards, 

on behalf of

Dr. Fernando C. Wehrmeister 

Academic Editor

PLOS ONE